# The Consequences of General Medication Beliefs Measured by the Beliefs about Medicine Questionnaire on Medication Adherence: A Systematic Review

**DOI:** 10.3390/pharmacy8030147

**Published:** 2020-08-17

**Authors:** Wejdan Shahin, Gerard A. Kennedy, Ieva Stupans

**Affiliations:** 1School of Health and Biomedical Sciences, RMIT University, PO Box 71, Bundoora, VIC 3083, Australia; Ieva.stupans@rmit.edu.au; 2School of Science, Psychology and Sport, Federation University, Ballarat, VIC 3083, Australia; g.kennedy@federation.edu.au

**Keywords:** medication adherence, general beliefs about medicine, hypertension, diabetes mellitus, asthma, chronic obstructive pulmonary disease

## Abstract

(1) Background: Medication adherence is a key determinant of patient health outcomes in chronic illnesses. However, adherence to long-term therapy remains poor. General beliefs about medicine are considered factors influencing medication adherence. It is essential to address the gap in the literature regarding understanding the impact of general beliefs about medicine on medication adherence to promote adherence in chronic illnesses. (2) Methods: PubMed, CINHAL, and EMBASE databases were searched. Studies were included if they examined medication beliefs using the Beliefs about Medicine Questionnaire in one of four chronic illnesses: hypertension, diabetes, chronic obstructive pulmonary disease, and/or asthma. (3) Results: From 1799 articles obtained by the search, only 11 met the inclusion criteria. Hypertension and diabetes represented 91% of included studies, while asthma represented 9%. Higher medication adherence was associated with negative general medication beliefs; 65% of the included studies found a negative association between harm beliefs and adherence, while 30% of studies found a negative association with overuse beliefs. (4) Conclusions: This review evaluated the impact of harm and overuse beliefs about medicines on medication adherence, highlighting the gap in literature regarding the impact of harm and overuse beliefs on adherence. Further research is needed to fully identify the association between general beliefs and medication adherence in people with different cultural backgrounds, and to explore these beliefs in patients diagnosed with chronic illnesses, particularly asthma and chronic obstructive pulmonary disease (COPD). Healthcare providers need to be aware of the impact of patients’ cultural backgrounds on general medication beliefs and adherence.

## 1. Introduction

Chronic illnesses are the main cause of death and disability globally, and are often discussed in terms of four major disease groups—cardiovascular diseases, cancers, chronic obstructive pulmonary disease (COPD), and diabetes mellitus type 2 (DMT2) [1]. Taking medications may be an essential part of chronic illness management. However, adherence to long-term therapy remains poor [2].

The World Health Organization defines adherence as “the extent to which a person’s behaviour of taking medication corresponds with agreed recommendations from a healthcare provider” ([3], p. 3). Several key factors have been suggested to influence adherence, including therapy management, the healthcare system, as well as individual patient factors [4]. Medication non-adherence can be classified as intentional or unintentional according to the patient’s perspective. Unintentional non-adherence refers to unplanned behavior; it is passive rather than active [5]. It sometime occurs when the patient wants to adhere, but is unable to because they lack capacity or resources. For example, they may not have understood the instructions, cannot afford co-payment costs, or find it difficult to schedule, administer, or remember the treatment [6]. Intentional non-adherence is considered a process in which the patient actively decides not to take medication or follow treatment recommendations, presumably having weighed the costs and benefits of treatment [5]. Medication non-adherence is best understood in terms of the perceptual factors (e.g., beliefs and preferences) influencing motivation to start and continue with treatment [6]. These perceptions are predominantly represented by beliefs about medicines as well as beliefs about the illness that the medication is intended to treat or prevent [7].

Different tools are used to monitor medication adherence, including self-monitoring or medication event monitoring systems. Regardless of medication adherence monitoring methods, abundant evidence reveals that medication adherence is suboptimal across diseases and different cultural background groups [8]. Adherence tends to be higher in diseases with greater perceived threat to health, such as HIV/AIDS and cancer, and lower for chronic conditions (e.g., asthma, hypertension, chronic obstructive pulmonary disease, and diabetes) [8]. Levels of adherence to DMT2 and hypertension (HTN) treatment regimens vary widely with estimates from 36% to 93% for DMT2, and 30% to 70% for HTN [9,10]. Up to 50% of patients diagnosed with COPD fail to take medications as directed and many do not use inhalers effectively [11]. It is also estimated that 30–70% of patients living with asthma are not adherent to preventative medications [12]. This problem may potentially be addressed by having a better understanding of people’s beliefs about medicines in general contexts.

The most widely used tool to elicit medication beliefs is the Beliefs about Medicines Questionnaire (BMQ) [13]. This questionnaire was developed by Horne and colleagues as a method for assessing cognitive representations of medication and was validated for use in patients suffering from common chronic diseases [14]. It has two major sections: specific and general. The specific section assesses patients’ beliefs about medications prescribed for a particular illness and consists of two scales that assess personal beliefs about the necessity of taking medication and concerns of having side effects of prescribed medication. The general section deals with more general beliefs about medicines and consists of two scales, the general-overuse scale which addresses views about the way in which medicines are used by medical practitioners, and the general-harm scale which assesses beliefs about the degree to which patients perceive medicines as essentially harmful and/or poisonous [7]. A recent meta-analysis [6] has described how specific beliefs about medicines determined through the BMQ are correlated to medication adherence; however, this meta-analysis did not evaluate the general section of BMQ.

Recent research suggests that general beliefs seem to be particularly useful in distinguishing peoples’ ‘orientation’ towards medicines in general [15]. Some people have fairly negative perceptions of modern medicines, which are often associated with the notion that the chemicals or unnatural origins of medicines are dangerous and that complementary treatments are more ‘natural’ and therefore safer [15]. Thus, these beliefs influence the patient’s initial orientation toward medicine adherence behavior and are likely to be strongly related to personal views about the specific prescribed medication [7].

When patients decide whether or not to use prescribed medicines, they are influenced not only by their views on medicines, but also by their self-perceived sensitivity to the medicines’ effects and their beliefs about whether the medicines might do harm or good. Therefore, patients might prefer non-conventional or complementary and alternative medicines such as herbal remedies rather than medications prescribed by their doctors [16]. This may lead to suboptimal levels of adherence and treatment. Thus, it is important to evaluate the impact of general beliefs about medicine on medication adherence as these beliefs may influence treatment preferences, pathways to care, and adherence to medication [15]. The use of herbal remedies and their acceptability as part of the treatment possibilities is viewed differently between countries. Such differences may affect beliefs about herbal remedies and their effects in relation to conventional medicines [16].

Although a number of systematic reviews have been conducted that aimed to identify factors that can influence medication adherence, including illness perceptions, specific medication beliefs, as well as culture and the demographic factors [6,17,18,19], none of these have explored the relationship between general beliefs about medicine determined through the BMQ and medication adherence of patients with chronic diseases such as HTN, COPD, asthma, and DMT2.

This systematic review aims to evaluate the impact of general medication beliefs on medication adherence in patients diagnosed with the most common chronic diseases in terms of population health; HTN, asthma, DMT2, and COPD.

## 2. Materials and Methods

The Preferred Reporting Items for Systematic Reviews and Meta-Analyses (PRISMA) guidelines were used to guide the review [20]. The search included articles retrieved from PubMed, CINAHL, and EMBASE, published from 1999 to the end of December 2019. The search commenced in 1999, as the BMQ was published in this year. Outlined in Table 1 is the combination of search terms that were used to source the relevant literature.

### 2.1. Inclusion Criteria

Articles were included if they (1) were published in peer reviewed journals; (2) used experimental studies (descriptive or observational analytic design); (3) were written in English; (4) pertained to one of the four chronic illnesses: DMT2, HTN, COPD, or asthma; (5) measured general medication beliefs using the BMQ; (6) included a clear description of the method of how medication adherence and general beliefs were measured; and (7) addressed the association between general beliefs and medication adherence.

### 2.2. Exclusion Criteria

Articles were excluded if they (1) were not peer reviewed articles; (2) focused on illnesses other than HTN, DMT2, COPD, or asthma; (3) assessed BMQ specific beliefs other than general beliefs; (4) measured general medication beliefs using different techniques other than BMQ; (5) included traditional medicine instead of prescribed medications; and (6) did not include adults.

### 2.3. Study Selection

After removing duplicates and including the titles and abstracts of the articles that met the inclusion criteria, the selection was screened by one researcher (W.S.). Next, copies of the full-text papers deemed potentially relevant by the first screening were subsequently independently fully analyzed by two researchers (W.S. and I.S.). Any remaining disagreements were discussed and a consensus was reached with a third researcher (G.K.).

### 2.4. Data Extraction Process

The data extracted from the included articles were synthesized. This process was carried out by two researchers (W.S., I.S.). The following data were extracted: demographic characteristics, the tools used to measure medication adherence, and a summary of findings of the studies which examined the association between general beliefs and medication adherence.

### 2.5. Risk of Bias in Individual Studies

Each eligible study included in this review was assessed using the Mixed Methods Appraisal Tool (MMAT) version 2018 [21], which comprises different methodological criteria depending on the type of the study. For cohort studies included in this review, the following five components were assessed: (1) if there was a match between respondents and the target population; (2) the variables were clearly defined and accurately measured; (3) the outcome data were completed; (4) the confounders were controlled in the design and analysis; and (5) during the study period, the intervention/exposure was administered as intended. For quantitative descriptive studies, the following five criteria were assessed: (1) if the way the sample was selected was relevant to address the research question; (2) if the sample was representative of the target population; (3) if the measurements were appropriate; (4) if the risk on non-response bias was low; and (5) the statistical analysis was appropriate to answer the research question. The benefit of this appraisal tool is that it can be used to assess all research designs through the use of a quality scoring system [22]. Two researchers (W.S., I.S.) independently scored articles, and scores were compared to identify differences, which were resolved through a discussion. Papers in this review were assigned a score based on the percentage of criteria that was met for the relevant study design. Papers were then categorized into one of three categories, where low quality was considered to be a score between 0% and 40%, medium quality between 41% 70%, and high quality between 71% and 100%.

## 3. Results

### 3.1. Article Selection

Figure 1 shows the process and results of the systematic search. The electronic search yielded 1799 articles. Four articles were found through reference lists and hand searching, and duplicates (n = 703) were removed. After that, the titles and the abstracts of the remaining articles were screened. One hundred and twenty-five articles were considered directly related to the aims of this review. A further comprehensive analysis of the full text articles resulted in an elimination of additional 114 articles. Finally, a set of 11 articles that met the inclusion criteria were selected.

### 3.2. Risk of Bias across Studies

Findings for a risk of bias across the quantitative descriptive studies are presented in Table 2. All papers were of medium or high quality. Studies included provided a detailed and clear definition of the inclusion criteria and described the settings clearly. Moreover, all the included studies used appropriate statistical analyses. The exposure and the outcome variables in all studies were measured using reliable and valid questionnaires (Cronbach’s alpha α = 0.5–0.84). A generalization of the results has been described in six of ten articles as one of the possible limitations, despite using convenience samples in all included studies. Two studies reported the possibility of generalizing the findings, as the analyzed sample matched the representative sample. In the included cohort study [23], confounding factors were identified and addressed, the intervention/exposure was administered as intended, and medication beliefs and adherence were clearly defined and accurately measured. However, the generalization of the results was described as one of the limitations of this article.

### 3.3. Tools Used to Measure Medication Adherence

Table 3 reports the tools used to measure medication adherence. Of the 11 studies, four studies measured medication adherence using the Medication Adherence Report Scale, while three studies used eight items of the Morisky Medication Adherence Scale (MMAS-8), and another three studies used four items of the Morisky Medication Adherence Scale (MMAS-4). Only one study assessed medication adherence using the Medication Adherence Questionnaire, and one used the Rief Adherence Index (RAI).

### 3.4. Findings of Individual Studies

Findings of the 11 reviewed studies are shown in Table 3. The studies appeared from 1999 to 2020, ten were cross-sectional studies that included surveys or interviews, while only one was a cohort study [23]. Most of the studies focused on patients with HTN (5 of 11) or DMT2 (5 of 11). Only one article focused on patients diagnosed with asthma [23]. The sample sizes ranged from 109 to 900 participants. Two of the studies were conducted in Germany, another three in the Middle East, two in the United States of America, and one each in China, United Kingdom, Nigeria, and Sweden. Participants in the included studies were recruited from various healthcare centers, clinics, hospitals, and adult day care settings. The age range of populations was 30–60 years old, and the median percentage of men included in the studies was 51.9%.

The influence of general beliefs about medicine on medication adherence was assessed in all 11 studies. Two of these evaluated general beliefs on medication adherence were assessed as one factor without measuring harm or overuse beliefs. In both of these studies, a negative association with medication adherence was reported [24,25].

#### 3.4.1. General-Harm

Nine studies evaluated the role of general-harm beliefs on medication adherence [2,23,26,27,28,29,30,31]. Three studies were carried out in the Middle East on patients diagnosed with hypertension and diabetes. These three studies found that participants who believed that medicines in general are intrinsically harmful substances also indicated lower medication adherence [2,26,27]. Harm beliefs about hypertensive and asthmatic medications were negatively associated with medication adherence in another two studies that were conducted in Germany [23,29]. Findings also were consistent with a cross-sectional study carried out in China that found a negative association between medication adherence and harm beliefs in patients with hypertension [30]. In all these studies, higher harm beliefs about the side effects of medications were associated with a lower medication adherence. Three studies reported no significant associations between harm beliefs and medication adherence [28,31,32] (Table 3).

#### 3.4.2. General-Overuse

Seven studies measured the impact of general-overuse beliefs on medication adherence [23,26,27,28,30,31,32]. Two studies conducted in Germany and China reported a negative association between medication adherence and beliefs about the overuse of asthmatic and hypertensive medications by medical practitioners [23,30]. In these studies, patients who had lower adherence levels were those who strongly considered that medications are overprescribed by their physicians, and held beliefs that if their physicians may had more time with them, they would prescribe fewer medicines [23,30]. Chinese patients in one of these two studies, reported a lack of trust in their doctors and other health professionals [30]. Five studies conducted in the UK, USA, and Middle East found no association between these beliefs and medication adherence [26,27,28,31,32] (Table 3).

## 4. Discussion

The effects of harm and overuse beliefs are considered by some health sociologists as hidden determinants of any treatment outcome and are considered critical predictors of medication adherence [32].

As shown in our results, a greater proportion of research papers included in this systematic review demonstrated a significant negative relationship between general-harm beliefs and adherence to medications. Two studies also reported a negative association between overuse beliefs and medication adherence in patients with the selected chronic conditions. However, some studies found no significant association with either general-harm or general-overuse beliefs or adherence. This may be attributed to a number of factors.

In this review, we identified significant associations of general-harm beliefs about medicines with medication adherence in studies that were conducted in Middle Eastern countries, China, and Germany [2,23,26,27,29,30]. Studies that found no association of general-harm beliefs with medication adherence were carried out in the UK, USA, and Sweden [28,31,32]. This suggests that certain cultural aspects might affect harm beliefs and thereby influence medication adherence.

A number of studies have shown that beliefs about medicines as harmful were positively associated with use of herbal remedies, but negatively with use of prescription medicines [16,33,34]. Patients from Asian cultural backgrounds expressed more negative views about medication than those from European cultural backgrounds. The exception for European backgrounds was that the use of herbal remedies was found to be very high in the general German population [35]. Patients from Asian cultural backgrounds were significantly more likely to perceive medicines as being intrinsically harmful, addictive substances and that the unnatural origins of medicines was dangerous and that complementary treatments are more ‘natural’ and therefore safer [15]. Patients from the Middle East preferred herbal supplements to conventional medicine and perceived no harm from using these supplements [36]. Previous research has reported that patients who attend herbal clinics were significantly more likely to perceive medicines as intrinsically harmful substances that are overused by doctors [37].

Different factors have been reported in literature that influence individuals’ preference to use natural remedies over their prescribed medications. These may include older age, having comorbidities, a lower educational level, and a lower socioeconomic status [38]. Therefore, it is imperative to take these factors into account when conducting future studies with regards to medication beliefs and adherence. Identifying barriers for each patient and adopting suitable techniques to overcome them will be necessary to improve medication adherence [39].

It was hypothesized that natural remedies used for chronic conditions were related to variability in medication use and adherence across different racial and ethnic groups. In addition, disclosure of natural remedies use during the medical encounter occurs less frequently among minority groups, which may be due to concerns about a negative response, or simply to practitioners not asking [8]. Identifying whether and how patients use natural remedies by healthcare providers is an important consideration in understanding adherence to medication among patients with chronic conditions.

This review also identified a negative association with overuse beliefs in only two studies. These beliefs if not alleviated would more likely hinder adherence to medications. It is widely reported that physician communication is key to improving treatment adherence and outcomes. Patient–healthcare provider communication helps the patients to form positive beliefs about treatment options. The patients who trust their healthcare provider, with the healthcare provider potentially showing friendliness, warmth, and concern, would not believe that the medication is overused [40]. Interpersonal communication has a significant association with the belief of medication overuse, i.e., an increase in interpersonal communication has been significantly associated with a decrease in medication overuse beliefs [40]. One of the studies that found a negative association between overuse beliefs and medication adherence was conducted in China [30]. Consistently, there is evidence in literature that patient–healthcare provider mistrust was considered as a major problem that has become a common part of the Chinese patient experience [41]. Patient trust may influence patients’ willingness to continue taking their medications and to stop perceiving their medications as overused [42]. One of the practical strategies to support this is to encourage healthcare providers to provide care that is concordant with the patient’s values, needs, and preferences, and that allows patients to provide their own input and participate actively in decisions regarding their medications and health [43]. When this occurs, the patient–doctor communication would help the patient to form positive beliefs about treatment options, thus enhance their medication adherence [40]. Medication beliefs are based on the individual’s education and knowledge, as well as the influence of healthcare providers, thereby giving rise to a mutual relationship between individuals’ beliefs and the environment to which they are exposed [14].

Previous research suggests that patients with low levels of education in comparison to those with high levels of education regard the effects of medications as harmful and believe that they are overused [44]. Participants in three studies that found no association of medication adherence with general medication beliefs reported good health literacy and high educational levels ranging from high school to college graduates [28,31,32]. Although education is an important factor, educational interventions alone represent a weak approach to enhance adherence to medication. A multidisciplinary approach is needed to enhance patient adherence to prescribed pharmacotherapy, a cornerstone of successful disease treatment [45]. To make this approach more effective, healthcare professionals, such as pharmacists, with their specific expertise and knowledge of medications, can have a very important role as patient educators and interventionists in various chronic conditions. Furthermore, pharmacists serve as an important link between patients and doctors [45].

The number of patients with chronic conditions is increasing dramatically and the real challenge is not only to identify non-adherent patients, but to prevent non-adherence behaviors to achieve better outcomes [46]. Patients’ initial decisions to not take medications might be influenced primarily by their general beliefs about medicine and consequently this will exacerbate concerns about the likely risk and the possible side effects of medications. These beliefs might outweigh the beliefs about potential benefits of medications, and thus patients intentionally tend not to adhere to their prescribed medications [7]. Patients who take medications for chronic conditions should be targeted through healthcare counselling, with the safety of their medications emphasized.

General medication beliefs, are likely subjected to modification with healthcare intervention and advice [37]. These beliefs are potentially formed from different personal characteristics, such as culture and religion [32]. It has been identified that patients’ cultural beliefs about medication-taking are factors contributing to intentional medication non-adherence [47]. Patients from different backgrounds may be reluctant to discuss medication beliefs, herbal therapies, home remedies, and religious practices with their healthcare providers. They often fear the practitioner’s disdain for these activities; therefore, they may withhold or alter this information to comply with the provider’s wishes [48]. However, healthcare providers should be encouraged to recognize gaps, confusions, and misconceptions about medications in patients from different cultures, and to provide sensitive care to populations from diverse ethnic backgrounds in order to achieve better medication adherence [18]. In addition, the identification of differences in beliefs about medication, between cultural groups, is essential for a greater understanding of the effects of cultural background on medicine-usage with potential implications for the conduct of prescribing-related consultations and for the provision of patient information on medication [15].

Our review is the first to show that general beliefs, particularly harm beliefs, are correlated with medication adherence in studies of some, but not all, populations. Therefore, we suggest using measures of these beliefs in future studies to predict medication adherence in different chronic illnesses.

There were some limitations in the current review, such as the risk for selection bias due to the possibility of missing some relevant studies because the research was limited to English language studies retrieved from three electronic databases. Another limitation was that the abstract screening was carried out by one researcher, thus there was no reliability check for abstract screening. Furthermore, the review was limited only to the common four chronic illnesses. A strength of the present review is the inclusion of the impact of general beliefs, which has not been mentioned before in any review about medication beliefs.

## 5. Conclusions

This review demonstrated a significant negative association between general beliefs about medication and medication adherence. Overuse and harm beliefs impact the actions of the individual and negatively affect medication-taking behaviors.

This review is the first to draw attention to enhancing medication adherence amongst chronically ill patients by changing their general beliefs about medication. This review also gives an insight to the need for future studies related to the impact of general beliefs on medication adherence. In addition, our study reveals the need for further studies about the impact of general beliefs about medications on medication adherence. In addition, this review highlights the gap in literature regarding studies that have investigated the role of medication beliefs on medication adherence in patients diagnosed with asthma and/or COPD.

## Figures and Tables

**Figure 1 pharmacy-08-00147-f001:**
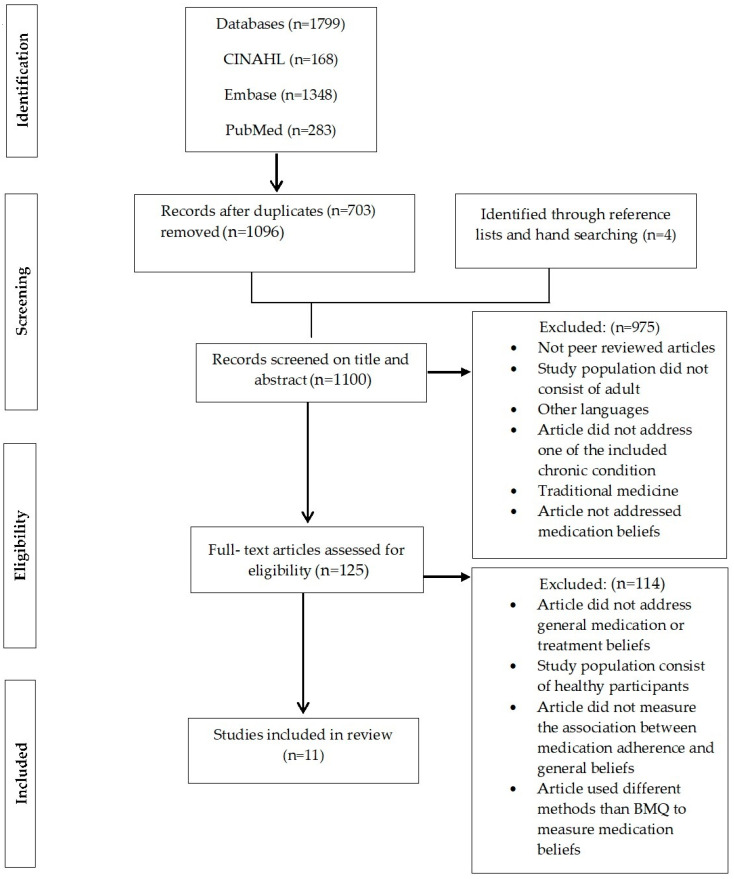
Preferred Reporting Items for Systematic Reviews and Meta-Analyses (PRISMA) flow chart of the literature search and study selection.

**Table 1 pharmacy-08-00147-t001:** Review search terms and databases reviewed.

Database	Search Terms
PubMed	(((((hypertension) OR (diabetes mellitus)) OR (asthma)) OR (chronic obstructive pulmonary disease)) AND (((((overuse beliefs) OR (harm beliefs)) OR (beliefs about medicine questionnaire)) OR (medication beliefs)) OR (general beliefs))) AND ((medication compliance [MeSH Terms]) OR (medication adherence [MeSH Terms]))
CINAHL with full text	beliefs about medicine questionnaire OR medication beliefs OR therapeutic beliefs OR overuse beliefs OR harm beliefs AND ((drug OR therapeutic OR medication)) AND (compliance OR adherence OR persistence)) AND ((chronic obstructive pulmonary disease) OR diabetes OR hypertension OR asthma)
EMBASE	‘ ‘medication compliance’ OR ‘medication adherence’ OR ‘therapeutic adherence’ OR (therapeutic AND (‘adherence’/exp OR adherence)) OR ‘treatment adherence’/exp OR ‘treatment adherence’ OR ((‘treatment’/exp OR treatment) AND (‘adherence’/exp OR adherence)) OR ‘medication persistence’ OR ‘medication’/exp OR medication OR ‘persistence’/exp OR persistence AND ‘treatment beliefs’ OR ((‘treatment’/exp OR treatment) AND (‘beliefs’/exp OR beliefs)) OR ‘medication beliefs’ OR ((‘medication’/exp OR medication) AND (‘beliefs’/exp OR beliefs)) OR ‘therapeutic beliefs’ OR (therapeutic AND (‘beliefs’/exp OR beliefs)) OR ‘beliefs about medicines questionnaire’/exp OR ‘beliefs about medicines questionnaire’ OR ‘general beliefs’ OR (general AND (‘beliefs’/exp OR beliefs)) OR ‘harm beliefs’ OR (harm AND (‘beliefs’/exp OR beliefs)) OR ‘overuse beliefs’ OR (overuse AND (‘beliefs’/exp OR beliefs)) AND (‘hypertension’/exp OR hypertension OR ‘chronic obstructive pulmonary disease’/exp OR ‘chronic obstructive pulmonary disease’ OR (chronic AND obstructive AND pulmonary AND (‘disease’/exp OR disease)) OR ‘asthma’/exp OR asthma OR ‘diabetes’/exp OR diabetes)

**Table 2 pharmacy-08-00147-t002:** Risk of bias in individual studies.

Methodological Quality Criteria	Lemay et al. [2]	Pereira et al. [24]	Olorunfemi and Ojewole [25]	Yousefabadi et al. [26]	Sweileh et al. [27]	Ross, Walker and MacLeod [28]	Wilhelm, Rief, and Doering [29]	Wei et al. [30]	Rajpura and Nayak [31]	Al-Ruthia et al. [32]
Is the sampling strategy relevant to address the research question?	Yes	Yes	Yes	Yes	Yes	Yes	Yes	Yes	Yes	Yes
Is the sample representative of the target population?	Unknown	No	No	Unknown	No	Yes	No	Yes	No	No
Are the measurements appropriate?	Yes	Yes	Yes	Yes	Yes	Yes	Yes	Yes	Yes	Yes
Is the risk of non-response bias low?	Yes	Yes	Unknown	Unknown	Unknown	Yes	Yes	Yes	Yes	Yes
Is the statistical analysis appropriate to answer the research question?	Yes	Yes	Yes	Yes	Yes	Yes	Yes	Yes	Yes	Yes
Total score	75% (High quality)	75% (High quality)	50% (Medium quality)	50% (Medium quality)	50% (Medium quality)	100% (High quality)	75% (High quality)	100% (High quality)	75% (High quality)	75% (High quality)

**Table 3 pharmacy-08-00147-t003:** Measurement tools, demographics characteristics, and findings of included studies.

Reference	Chronic Condition	Demographics Characteristics			Duration of Illness	Measurement of Medication Adherence	Association with Medication Adherence	Statistical Analysis
		Age Mean (SD)	Male N (%)	Settings				
[30]	Diabetes Mellitus type 2	62.5 (13.9)	174 (55.2)	Two large urban hospitals in Hefei and Tianjin, China	5 years	Medication Adherence Report Scale	- Significant positive correlation between overuse and medication non-adherence	r = 0.49, *p* < 0.01
							- Significant positive correlation between harm and medication non-adherence	r = 0.49, *p* < 0.01
[24]	Diabetes Mellitus type 2	59.2 years	225 (58.1)	40 healthcare centers of the northern region of Portugal.	No more than 1 year	Medication Adherence Scale	- Significant negative correlation between general beliefs and medication adherence	r = −0.13, *p* < 0.01
[29]	Hypertension	53.93 (15.46)	133 (51.3)	Link to the survey was spread via various mail distributors (e.g., the German Society for Hypertension). The link was also printed on flyers that were distributed to pharmacies and hospitals in Germany.	9 years	Rief adherence index (RAI)	- Significant negative correlation between harm beliefs and medication adherence	β = 0.61, *p* < 0.001
[32]	Hypertension	60 years	43 (22.6)	Seniors centers in Memphis (USA)	-	Morisky Medication Adherence Scale (MMAS-8)	- No significant correlation between overuse beliefs and medication adherence	*p* = 0.167
							- No significant correlation between harm beliefs and medication adherence	*p* = 0.323
[27]	Diabetes Mellitus type 2	58.3 (10.4)	189 (46.7)	Al-Makhfia governmental diabetes primary healthcare clinic in Nablus, Palestine	8.5	Morisky Medication Adherence Scale (MMAS-8)	- No significant correlation between overuse beliefs and medication adherence	OR, 1.0 95% CI (0.94–1.1) *p* = 0.95
							- Significant negative correlation between harm beliefs and medication adherence	OR, 1.1 95% CI (1.1–1.2) *p* < 0.001
[31]	Hypertension	>55 years	75 (64.1)	Adult day care center in New York City (USA)	>7 years	Morisky Medication Adherence Scale (MMAS-4)	- No significant correlation between overuse beliefs and medication adherence	β = −0.074, *p* = 0.473
							- No significant correlation between harm beliefs and medication adherence	β = −0.071, *p* = 0.396
[2]	Cardiovascular, Diabetes	>18 years	444 (56.7)	Primary healthcare, Kuwait	-	Medication Adherence Report Scale	- Significant negative association between harm beliefs and medication adherence	β = −0.46, *p* < 0.05
[25]	Diabetes Mellitus type 2	>27 years	80 (44.4)	Diabetic clinic in three hospitals in Benin-city, Nigeria	-	Morisky Medication Adherence Scale (MMAS-4)	- Significant negative association between general beliefs about medication and medication adherence	r = −0.208 *p* = 0.005
[28]	Hypertension	59.92 (12.16)	267 (51.9)	A secondary care hypertension clinic and shared care scheme, UK	-	Morisky Medication Adherence Scale (MMAS-4)	- No significant correlation between harm beliefs and medication adherence	*p* > 0.05
							- No significant correlation between overuse beliefs and medication adherence	*p* > 0.05
[23]	Asthma	56.7 (15.9)	203 (50.5)	Different hospitals, primary care, and specialists practices in the area of Regensburg, Germany	9.94	Medication Adherence Report Scale	- Significant negative correlation between harm beliefs and medication adherence in asthmatic patients	OR, 0.5 95% CI [0.3–1.03], *p* = 0.02
							- Significant negative correlation between overuse beliefs and medication adherence in asthmatic patients.	OR, 0.56 95% CI [0.34–0.93] *p* < 0.01
[26]	Hypertension	>30 years	251 (56.3)	All health centers and clinics of internal and heart diseases under coverage of Zahedan University of Medical Sciences based in Zahedan, Iran	<15 years	Medication Adherence Questionnaire	- Significant negative association between harm beliefs and medication adherence	*p* = 0.02
							- No significant association between overuse beliefs and medication adherence.	*p* = 0.1

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
