# Peer review of "The Consequences of General Medication Beliefs Measured by the Beliefs about Medicine Questionnaire on Medication Adherence: A Systematic Review"

_pharmacy, 2020, doi:10.3390/pharmacy8030147_

Round 1
Reviewer 1 Report
This manuscript consists of a systematic review aiming to evaluate the impact of general medication beliefs on medication adherence in patients diagnosed with chronic diseases such as arterial hypertension (HTN), chronic obstructive pulmonary disease (COPD), asthma and diabetes mellitus type 2 (DMT2). The relationship between general beliefs about medicines and medication adherence was determined through the Beliefs about Medicines Questionnaire (BMQ).
The design, methods and results are clearly presented. By employing the well-established PRISMA guidelines, with clearly defined inclusion and exclusion criteria, data extraction process and risk of bias in individual studies analysis, authors figured out that a significant negative association between harm and overuse beliefs about medication and medication adherence exists in some populations, which supports the speculation of a possible significant impact of patients’ cultural backgrounds on general medication beliefs and adherence.
Despite several limitations well documented by authors in the “Discussion” section, and although this systematic review does not bring novelties of greater, its results may add interesting information to some medication adherence studies.
There are, however, some points the authors should address:
1) Page 2, line 84
“The search included articles retrieved from PubMed, CINAHL, and Embase, published from 1980 to end of December 2019.”
I wonder why authors searched from 1980, if the BMQ was only published in 1999.
That statement from authors does not seem to be in accordance with the other one made in page 5, line138 “The 11 studies reviewed, and shown in Table 2, appeared from 1999-2020…”
2) 3.1. Article selection
The values mentioned in the text do not seem to be in accordance with those presented in figure 1.
“The electronic search yielded 2460 articles.”
Figure 1 mentions 1799 articles.
“One hundred and seventeen articles were considered directly related to the aims of this review.”
Figure 1 mentions 125 articles.
“A further comprehensive analysis of the full text articles resulted in elimination of additional 106 articles.”
Figure 1 mentions 114 articles.
Please clarify.
3) The meaning of the abbreviation used in Figure 1 (BMQ) should be explained in the figure legend, given that, according to international guidelines, the reader must be able to read each figure / table independently.
4) Page 8, last line
Where it reads:
“Significant negative association between harm and medication beliefs”
should read
“Significant negative association between harm and medication adherence”
5) There are also some specific grammatical issues that must be corrected.
Example:
In addition, previous research reported that patients who attending a herbal clinic…
Reviewer 2 Report
This is a potentially interesting paper, particularly regarding differences between cultures in terms of predictors of adherence. A number of issues need to be addressed before publication.
Abstract. The specific research gap that is being addressed needs to be clearer in the abstract. The conclusion of the abstract then needs to follow logically from the results presented in the abstract.
Background: The difference between specific and general beliefs needs to be clearer. That way the reader may understand better why it is important to address general as well as specific beliefs and therefore why this review is important. In addition, please explain why you did not include cancer in your review as you also i this is one of the most common chronic conditions.
Methods: Was there a reliability check for abstract screening? If yes, please describe. If not, this needs to be added as a limitation. How many reviewers carried out data extraction and assessment of bias? The tool used for assessing risk of bias may not be appropriate as there was no exposure being measured in this review. Table 2 contains results and should not therefore be referenced in the methods section.
Discussion: Be careful not to present new results in the discussion section e.g. lines 210-212. Also be careful of recommending counseling as a catch all solution to addressing adherence issues and medication beliefs.
General: There are a number of typos and sentences that are not clear. Please proofread carefully.
Round 2
Reviewer 2 Report
Thank-you for making the suggested changes to the paper, it is is now much improved.
The abstract is also improved but still requires further work. The gap being filled by the study needs to be clear by the of the background section. The results presented in the abstract do not support the conclusion reached in the abstract.
Author Response
Thank you for the comments.
Reviewer comment 1:
The gap being filled by the study needs to be clear by the of the background section.
Response:
We have included the following to the background:
"It is essential to address the gap in the literature regarding understanding the impact of general beliefs about medicine on medication adherence to promote adherence in chronic illnesses."
Reviewer comment 2:
The results presented in the abstract do not support the conclusion reached in the abstract.
Response:
We have included the following:
Results:
"65% of the included studies found a negative association between harm beliefs and adherence, while 30% of studies found a negative association with overuse beliefs"
Conclusion:
"highlighting the gap in the literature regarding the impact of harm and overuse beliefs on adherence".